# Meteorin-like is an injectable peptide that can enhance regeneration in aged muscle through immune-driven fibro/adipogenic progenitor signaling

David E. Lee[1,2], Lauren K. McKay[2,3], Akshay Bareja[1,2], Yongwu Li[4], Alastair Khodabukus[4], Nenad Bursac [4], Gregory A. Taylor[5,6,7,8,9], Gurpreet S. Baht[2,10] & James P. White [1,2,9] ✉

Pathologies associated with sarcopenia include decline in muscular strength, lean mass and regenerative capacity. Despite the substantial impact on quality of life, no pharmacological therapeutics are available to counteract the age-associated decline in functional capacity and/or, resilience. Evidence suggests immune-secreted cytokines can improve muscle regeneration, a strategy which we leverage in this study by rescuing the age-related deficiency in Meteorin-like through several in vivo add-back models. Notably, the intra-muscular, peptide injection of recombinant METRNL was sufficient to improve muscle regeneration in aging. Using ex vivo media exchange and in vivo TNF inhibition, we demonstrate a mechanism of METRNL action during regeneration, showing it counteracts a pro-fibrotic gene program by triggering TNFα-induced apoptosis of fibro/adipogenic progenitor cells. These findings demonstrate therapeutic applications for METRNL to improve aged muscle, and show Fibro/Adipogenic Progenitors are viable therapeutic targets to counteract age-related loss in muscle resilience.

Sarcopenia and the age-related decline in muscular strength and regenerative capacity contribute directly to the loss of autonomy, greater risk for hospitalization, and healthcare burden[1,2]. Improving resilience could allow for improved mobility and independence. Considering skeletal muscle accounts for nearly half our body mass and is susceptible to functional loss through injury, surgical trauma, or pathology, therapeutic options are greatly needed. Although lifestyle interventions (i.e., exercise or dietary restriction) have been effective in improving muscle resilience in aging[3,4], there are no available pharmacological therapies to improve muscle repair or to counteract the reduction in muscle resilience in the context of aging. Our group has recently reported the protein Meteorin-like (Metrnl) to have robust effects on immune function following muscle damage[5]. We propose that Metrnl can be leveraged as a pharmaceutical therapy to enhance

[1]Department of Medicine, Division of Hematology, Duke University School of Medicine, Durham, NC 27710, USA. [2]Duke Molecular Physiology Institute, Duke University School of Medicine, Durham, NC 27701, USA. [3]Division of Oral and Craniofacial Health Sciences, Adams School of Dentistry, University of North Carolina at Chapel Hill, Chapel Hill, NC, USA. [4]Department of Biomedical Engineering, Duke University, Durham, NC 27710, USA. [5]Geriatric Research, Education, and Clinical Center, VA Health Care System, Durham, NC 27701, USA. [6]Department of Medicine, Division of Geriatrics, Duke University School of Medicine, Durham, NC 27710, USA. [7]Department of Molecular Genetics and Microbiology, Duke University School of Medicine, Durham, NC 27710, USA. [8]Department of Immunology, Duke University School of Medicine, Durham, NC 27710, USA. [9]Duke Center for the Study of Aging and Human Development, Duke University School of Medicine, Durham, NC 27701, USA. [10]Department of Orthopaedic Surgery, Duke University School of Medicine, Durham, NC 27710, USA. ✉e-mail: james.white@duke.edu

aged muscle regeneration by positively modulating the cellular environment throughout the regenerative process.

Age-related regenerative impairments are characterized by delayed resolution of injury, incomplete restoration of contractile components and appearance of fibrotic, non-contractile tissue within the muscle. Factors contributing to incomplete regeneration include cell-based[6] detriments, changes in the local tissue environment[7] as well as perturbations in systemic factors[8]. Infiltrating immune cells extravasate to damaged muscle and secrete various peptides and cytokines directly into the muscle milieu to drive the regenerative process. Leveraging circulating factors that contribute to the microenvironmental cues lost later in life has been shown to be a highly effective therapeutic strategy[8,9]. Pro-regenerative peptides secreted in the muscle micro-environment are especially attractive therapeutic options because of their biologic nature, stability, and ease of delivery to the site of injury.

Meteorin-like (Metrnl) was originally identified as a myokine, shown to regulate adipose tissue metabolic programming[10]. Since then, Metrnl has demonstrated a role in various cell types including alternatively activated macrophages[11], osteoblasts[12,13], and adipocytes[14]. We recently demonstrated the necessity for Metrnl for successful muscle regeneration in vivo[5]. Using cell-type specific deletion of *Metrnl*, we demonstrated that myofiber-specific expression of *Metrnl* was dispensable, while macrophages are a primary source of Metrnl that promotes the initial phases of the regenerative process[5]. The administration of recombinant Metrnl protein to genetically deficient mice at the site of injury showed efficacy as a therapeutic approach to promote

an anti-inflammatory/pro-regenerative environment within the muscle and enhance myogenesis[5]. The utility of Metrnl to improve immune regulation and environmental cues during regeneration leads to the hypothesis that Metrnl is deficient in models of pathological muscle regeneration, including aging, and therefore a likely therapeutic target to restore healthy tissue regeneration.

In this study, we first demonstrate a deficiency of Metrnl in aged muscles following injury, and then show in several models of in vivo muscle injury that aged mice can be rejuvenated by Metrnl add-back approaches. Using preclinical models, we demonstrate how intramuscular delivery of recombinant METRNL is sufficient to improve muscle regeneration in aging. We then demonstrate a mechanism of METRNL action during aged regeneration by showing that METRNL enhances the immune responsiveness to counteract a pro-fibrotic program through TNF-induced fibro/adipogenic progenitor (FAP) apoptosis.

## Results

### Reduction in Metrnl expression with age

Aged mice (24–27 months old) had a profound reduction in *Metrnl* expression across the recovery time course when compared to the young controls (Fig. 1a). Both age groups showed an induction of Metrnl expression during the initial days after BaCl$_2$-induced damage, eventually returning to pre-injury values after several weeks. Using a more physiological model of muscle damage (downhill treadmill running), we found a similar reduction of muscle *Metrnl* gene expression in aged mice (Fig. 1b) along with reductions in serum protein when

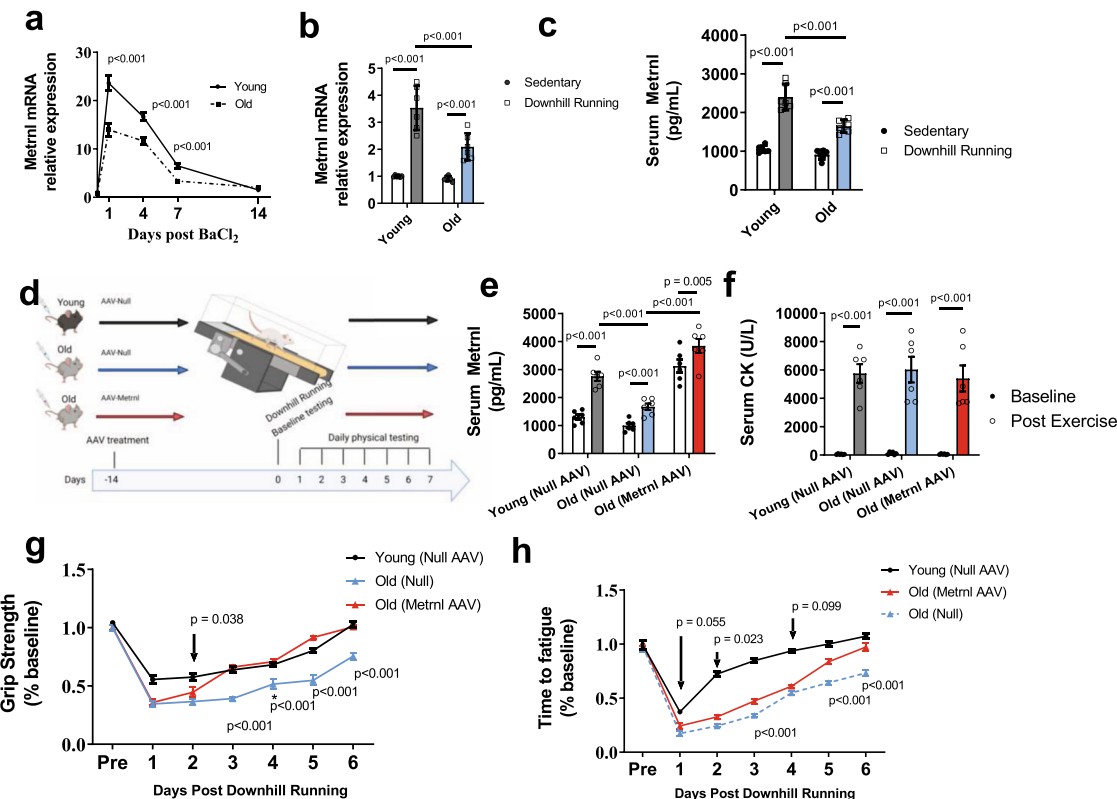

**Fig. 1 | Meteorin-like expression is reduced in aged muscle regeneration.**
**a** Metrnl mRNA measured by qPCR from mouse TA muscles following BaCl$_2$-induced muscle damage ($n = 6$ independent animals/age/timepoint); two-tailed $t$ test was used to compare young to old at each timepoint. **b** Metrnl mRNA and **c** plasma protein in young and old mice 24 h following downhill running-induced muscle damage ($n = 6$ independent animals/group); 2 × 2 ANOVA was used with Fisher's LSD post hoc analysis. **d** Graphic representation of mouse downhill running experiments. **e** Circulating METRNL before and at 24 h post downhill running ($n = 6$ independent animals/group); 2 × 3 ANOVA was used with Fisher's LSD post hoc

analysis. **f** Circulating creatine kinase measure before and 72 h after injury. $n = 6$ independent animals/group; 2 × 2 ANOVA was used with Fisher's LSD post hoc analysis. **g** Grip strength and **h** running time to fatigue as a percent of baseline measured before and each day following downhill running ($n = 6$ animals/group; two-tailed $t$ test was used to compare old(Null) to old(Metrnl) at each timepoint. Source Data are provided as a source data file. Graphics created with BioRender.com. All $p$ values are indicated with line connecting comparison groups. Data are presented as mean values ± SEM. WT wild-type, KO METRNL−/−, Y young, O old, ISO isochronic; HET heterochronic.

measured 24 h after damage induction (Fig. 1c). As we previously demonstrated the necessity for METRNL during muscle regeneration[5], we tested the hypothesis that enhancing the expression of *Metrnl* is sufficient to improve the age-related decline in muscle recovery. We administered AAV8-Null or AAV8-Metrnl via tail vein injection to mice 24 months old and performed exhaustive downhill running in order to induce muscle damage (Fig. 1d). Circulating METRNL protein confirmed successful AAV restoration in METRNL levels in response to damage (Fig. 1e). Circulating creatine kinase levels, a signature of muscle damage, was similar across all groups when measured immediately before and 72 h after downhill running (Fig. 1f). Functional measurements taken six days following damage show an age-related decline in the recovery of grip strength and running capacity when compared to young controls (Fig. 1g, h). Conversely, recovery was enhanced with AAV-mediated forced expression of METRNL in aged mice (Fig. 1g, h). These experiments demonstrate the sufficiency of METRNL to improve functional regeneration in aged mice by using a non-specific overexpression vector.

### Circulating immune cells contribute to reduced *Metrnl* expression in aged regeneration

Evidence suggests *Metrnl* expression is highly enriched in infiltrating monocyte and macrophage populations following muscle damage[5] so we hypothesized the age-related *Metrnl* insufficiency was specifically a result of reduced macrophage expression and secretion levels of Metrnl. We performed global transcriptomics analysis on bone marrow-derived macrophages (BMDM) from 4-month (young) and 24-month (old). *Metrnl* was among the many genes identified as differentially expressed at an adjusted *p* value below 0.05 between M0 macrophages of young and old BMDM (Fig. S1A, Table S1). We saw a similar decline in METRNL protein secretions and *Metrnl* mRNA levels in BMDM cultured in IL-4 or LPS-supplemented media (Fig. S1B, C) showing that the reduction in METRNL is consistent across several states of BMDM-derived macrophage polarization. To further verify expressions patterns of *Metrnl* to specific cells types, we performed in silico analysis of the previously published, large scale scRNA-seq integrated dataset published by McKellar et al.[15]. This integrated dataset includes ~365,000 cells including samples across the mouse lifespan (2 months through 30 months age) and across a time course of muscle injury (uninjured through 28 days post injury) with more details available in the original publication[15]. We found distinct patterns of *Metrnl* expression levels in clusters of cells identified along the myeloid lineages (monocytes, macrophages, dendritic cells; Fig. S1d, e). When analyzed with respect to time following injury and age of mouse, the pattern of *Metrnl* expression showed a similar dynamism in the immune cell clusters to that of our qPCR analysis on whole muscle homogenate (Fig. S1f, g). Because of this consistent pattern of myeloid-derived expression of *Metrnl* across multiple models, we tested the hypothesis that in vivo replacement of aged circulating leukocytes with young cells was sufficient to rejuvenate aged muscle regeneration. To test this hypothesis, we utilized two well-established models of heterochronic rejuvenation of circulating immune cell populations (parabiosis and bone marrow transplant). First, we used wild-type isochronic (old-old; ISO) or heterochronic (young-old; HET) parabiosis which allows a reproducible sharing of the circulation of mice. Once anastomosis was established (1 month of parabiosis), we induced muscle damage via intramuscular BaCl₂ injection to the aged parabiont and assessed regenerative capacity (Fig. 2a). Regenerating muscle was harvested 1 day and 14 days post injury and assessed for *Metrnl* expression (Fig. 2b) and CSA, respectively. Fourteen days following BaCl₂ injection, we saw a significant improvement in the regenerated muscle cross-sectional area (CSA) of the old-HET group compared to that of the old-ISO controls (Fig. 2c, d). The rescue of *Metrnl* mRNA and muscle CSA were each lost when the old-HET pairing included a young mouse with global deletion of the *Metrnl* gene (Metrnl−/−; Fig. 2a–d) demonstrating

the role of METRNL in the heterochronic improvements to muscle regeneration.

To substantiate these findings without the confounding factors associated with heterochronic parabiosis (changes in activity levels, epigenetic alteration in aged muscle), we repeated these experiments using bone marrow chimeras. We established bone marrow chimera in old (24 m), wild-type hosts reconstituted with either old WT, young WT, or young *Metrnl*−/− bone marrow (Fig. 2e). After subjecting the recipient chimera mice to BaCl₂-induced muscle damage, we found a similar pattern of bolstered *Metrnl* mRNA expression and improved restoration of muscle architecture as that of the parabiosis experiments (Fig. 2f–h). Once again, this effect was lost in aged mice receiving young *Metrnl*−/− chimera.

To ensure *Metrnl* deficiency was not altering hematopoietic cell lineages, we performed flow cytometry on peripheral blood and bone marrow from *Metrnl*−/− mice and saw no statistically significant changes in Ter119+ cells, Sca1+ cells, VEGF-R+ cells, CD45+ cells, or CD45−, CD34+ cells (Fig. S2a, b). These experiments demonstrate the capacity of young circulating immune cells to enhance muscle regeneration in aged mice and the dependence of this rejuvenation on *Metrnl* expression.

### Direct, intramuscular administration of recombinant METRNL as a regenerative therapeutic

We next sought to test the hypothesis that aged muscle regeneration could be improved by pharmacological replacement of METRNL by means of an injectable recombinant protein. Administration of recombinant rMETRNL via intramuscular (IM) injection was performed on mice following BaCl₂-induced muscle damage. The administration strategy was devised to mirror the natural *Metrnl* response to injury in healthy control muscle (Fig. 1a) which showed a robust *Metrnl* expression, early in the regenerative process. Using 4-month (young) healthy controls or 24-month-old mice, we injured tibialis anterior (TA) muscles by BaCl₂ injection followed by 3 consecutive treatments of PBS (control) or 0.1 ug rMETRNL immediately, 24 h and 48 h following BaCl₂ administration (Fig. 3a). Based on previous evidence of shifting macrophage cell profile following muscle injury in *Metrnl*−/− mice[5], we used flow cytometry to measure the proportion of macrophages that expressed Ly6C as a marker for inflammatory vs anti-inflammatory phenotypes[16]. Macrophages were determined by CD45 + , CD11b + , SiglecF/Ly6G-, F4/80+ cells (Fig. 2b). Twenty-Four hours following BaCl₂ injury, no differences were found in the number of Ly6C hi and low macrophages between groups (Fig. 3c). At 3 days post injury, Ly6C^hi cells constituted a greater proportion of macrophages in old muscle while there was a shift towards Ly6C^lo cells in young PBS-treated or old rMETRNL-treated muscle (Fig. 3d). By post injury day 5, the proportion of Ly6C^hi cells has predominantly shifted towards Ly6C^lo cells in all groups although the young group had lost nearly all of the remaining Ly6C^hi cells (Fig. 3e). This accelerated shift with rMETRNL administrations suggest an improved regenerative immune profile in aged mice. To assess this, we analyzed muscle histology at 14 days post injury, finding a greater average CSA of muscle fibers in old rMETRNL mice compared to old PBS control mice (Fig. 3f, g). Finally, to ensure the histological findings related to functional contractile strength, we performed in situ force measurements on the regenerating TA muscle. We show that old mice had significantly reduced absolute force and specific force generation compared to young mice; a difference that was corrected in the old rMETRNL group (Fig. 3h–i). Results from these experiments demonstrate a proof-of-concept that aged muscle regeneration can be enhanced by simple intramuscular administration of the recombinant rMETRNL protein.

### Transcriptomic rejuvenation of aged muscle regeneration by intramuscular injection of rMETRNL

To build upon this therapeutic model, we sought to identify a cellular mechanism to explain how rMETRNL alters the aged environment to

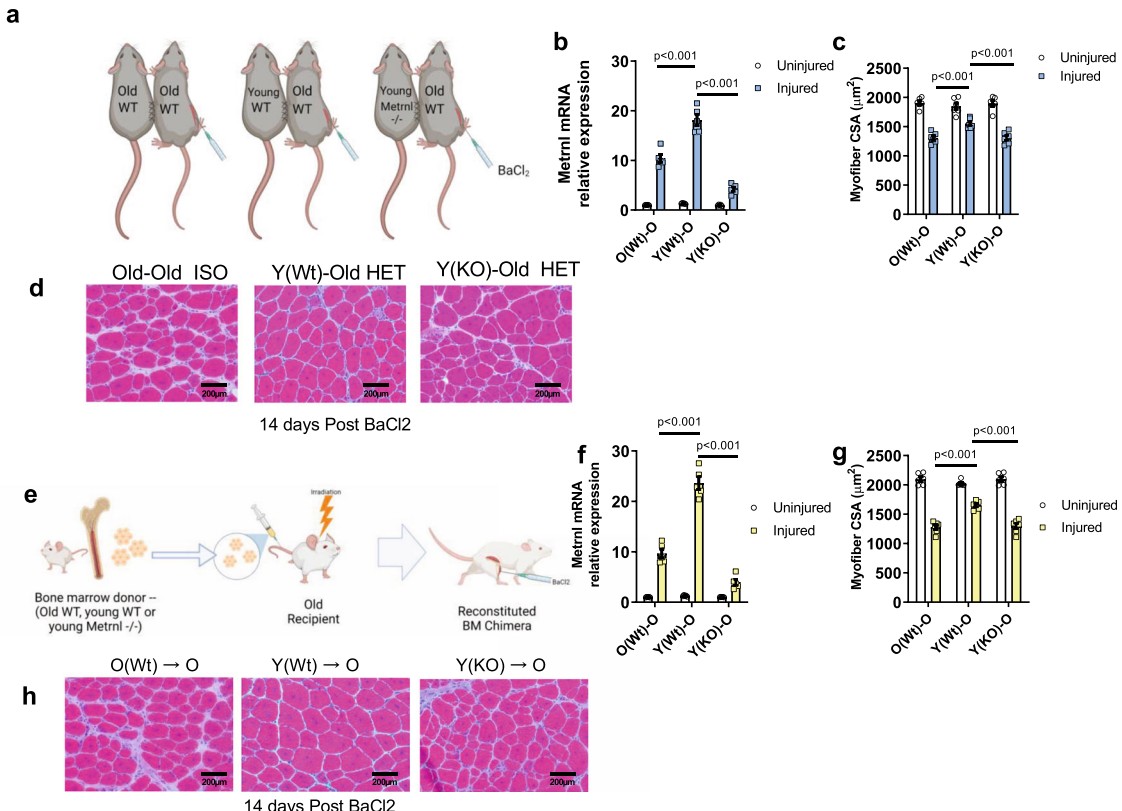

**Fig. 2 | Replacement of circulating Metrnl improves aged muscle regeneration following BaCl₂ injection. a** Graphical representation of Heterochronic parabiosis groups (*n* = 6/old WT parabionts tested per group). **b** *Metrnl* mRNA content measured by qPCR from uninjured and 24-hours post-injury TA muscles of old wild-type parabiont (*n* = 5 animals/group); 2 × 3 ANOVA was used with Fisher's LSD post hoc analysis only to compare differences between injured groups. **c** Cross-sectional area quantified from micrographs (*n* = 6 animals/group); 2 × 3 ANOVA was used with Fisher's LSD post hoc analysis only to compare differences between injured groups. **d** Representative Hematoxylin and Eosin-stained micrographs from **c**– 14 days following BaCl2-induced muscle damage in the old wild-type parabionts. **e** Graphical representation of Bone marrow chimera experiment. **f** *Metrnl* mRNA

content measured by qPCR from uninjured and 24-hours post-injury TA muscles of Old chimera (*n* = 5 animals/group); 2 × 3 ANOVA was used with Fisher's LSD post hoc analysis only to compare differences between injured groups. **g** Cross-sectional area quantified from micrographs (*n* = 6 animals/group); 2 × 3 ANOVA was used with Fisher's LSD post hoc analysis only to compare differences between injured groups. **h** Representative Hematoxylin and Eosin-stained micrographs of **g** 14 days following BaCl2-induced muscle damage in the Old chimera. Source Data are provided as a source data file. Graphics created with BioRender.com. All *p* values are indicated with lines connecting comparison groups. Data are presented as mean values ± SEM.

enhance muscle regeneration. We performed global transcriptomic analysis on whole muscle mRNA extracts. We chose 8 days post injury for this analysis to identify changes in gene expression signatures during the regenerative phase following the macrophage cellular shift[17] (Fig. 4a). Transcriptomic analyses measured 14,226 unique transcripts above background levels with 12,883 of these co-expressed among all sample groups (Fig. 4b). Principal component analysis showed similarities in gene expression signatures between mRNA taken from young PBS samples with that of old rMETRNL samples. These clusters both clearly distinguished from old PBS samples based on PC1 (38.55%) and PC2 (14.85%) together accounting for over 50% of variability among gene expression patterns, despite a high degree of variation seen within the old samples (Fig. 4c). We performed unsupervised, hierarchical clustering analysis on the 311 genes that were statistically different between young and old groups and visualized these results using heatmap and dendrogram plots, which show a consistent clustering young PBS and old rMETRNL separate from the old PBS (Fig. 4d). The 20 transcripts that were most significantly altered between Old PBS and Old rMETRNL groups by *p* value are shown in Fig. 4e–notably including multiple ECM-related genes (*Col24a1, Col3a1*) that are markers of fibrosis. We performed Gene Set Enrichment Analysis (GSEA) on lists of significantly altered genes using Gene Ontology (Biological Process–2018) and Reactome databases. GSEA revealed several pathways upregulated in old rMETRNL-treated

muscles compared to old PBS including extracellular matrix remodeling, organization and components from both databases (Fig. 4f). Results from this transcriptomic study were twofold: (1) a clear signature was present where old muscle showed upregulated pathways involved in collagen formation, ECM development and ECM component organization which was reversed by rMETRNL and (2) they substantiate an intramuscular METRNL injection following injury can rejuvenate the transcriptomic signature of aged muscle.

## Single-cell RNA sequencing inferential analysis of cell-to-cell communication

We next aimed to characterize potential cell-to-cell communication networks, which might explain how a transitioning macrophage phenotype contributes to altered fibrotic gene expression signatures. To test this, we performed single-cell RNA sequencing analysis on cells isolated from young PBS, old PBS and old rMETRNL mice 4 days post BaCl₂-induced damage (Fig. 5a). Following data normalization and transformation, 35,836 cells across all three samples (triplicate samples of each experimental group were pooled, isolated and submitted for sequencing) contained 18,266 features and were visualized using a 2D UMAP (Fig. S3A). UMAP clustering showed even distribution among each of the samples (Fig. S3B). Using canonical gene markers, ten distinct cell clusters were identified representing typical cell types seen during muscle regeneration (Fig. S3C, Fig. 5b). The two most abundant

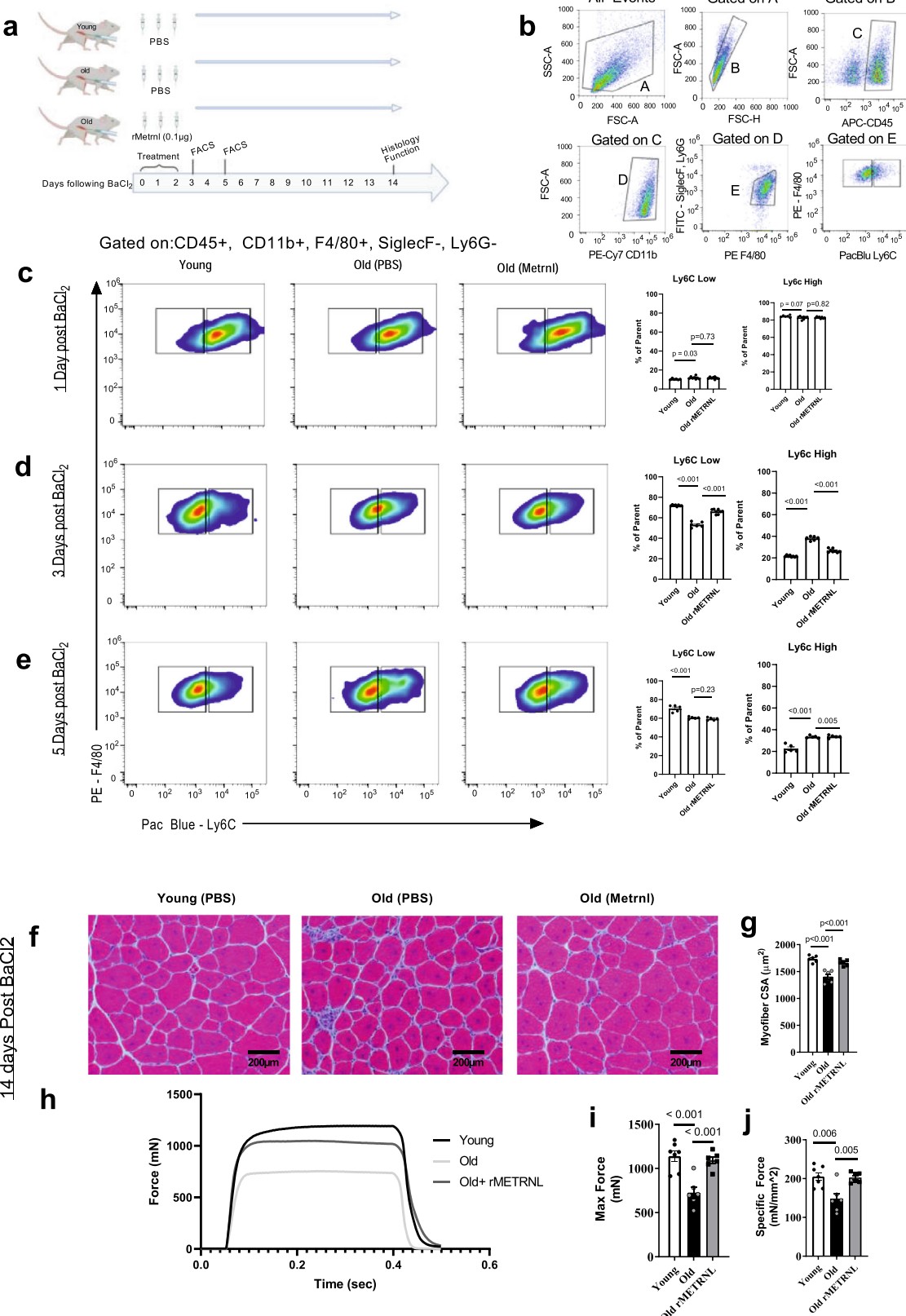

cell populations represented included CD52+ immune cells and Pdgfra + FAP cells, which are typical observations for 4 day-post injury timepoint (Fig. S3f, g). We performed cell-to-cell inferential analysis to identify ligands from a known secretory cell population to that of an unknown receptor-expressing cell population[18]. We used immune cells as the primary ligand source cell-type because of our previous findings

which demonstrated a shifting macrophage cell population in aged muscle treated with rMETRNL (see Fig. 3b, c). Results of cell-to-cell communication analysis revealed the greatest interaction strength and greatest number of interactions occurred between immune cells and the FAPs (Fig. 5c–e). When these receptor/ligand pairings were categorized by cell-to-cell communication pathways, the most significantly

**Fig. 3 | Intramuscular treatment with rMETRNL improves aged muscle regeneration through transition of the inflammatory/anti-inflammatory immune profile. a** Graphical representation of rMetrnl treatment groups and timeline for experiments. **b** Example gating strategy used for macrophage identification and subtype analysis. The final dot plot (F4/80 vs Ly6C) represents the data shown in **c**–**e**. Representative flow cytometry results and bar graphs of cells isolated from tibialis anterior muscles **c** 1 day (n = 6 young, 10 old, 10 old +rMetrnl), **d** 3 days (n = 8 young, 6 old, 8 old + rMetrnl), and **e** 5 days (n = 5 animals/group) post injury; one-way ANOVA was used with Fisher's LSD post hoc analysis. **f** Hematoxylin and Eosin-stained micrographs of tibialis anterior muscles 14 days follow BaCl₂-induced muscle damage and **g** quantified cross-sectional area (n = 6/group); one-way ANOVA was used with Fisher's LSD post hoc analysis. **h** Representative force generation following tetanic stimulation of tibialis anterior muscles in situ. Summary data (n = 7 young, 6 old, 6 old+rMetrnl animals) for maximum force (**i**) and specific force (**j**); one-way ANOVA was used with Fisher's LSD post hoc analysis. Source Data are provided as a source data file. All p values are indicated with line connecting comparison groups. Graphics created with BioRender.com. Data are presented as mean values ± SEM.

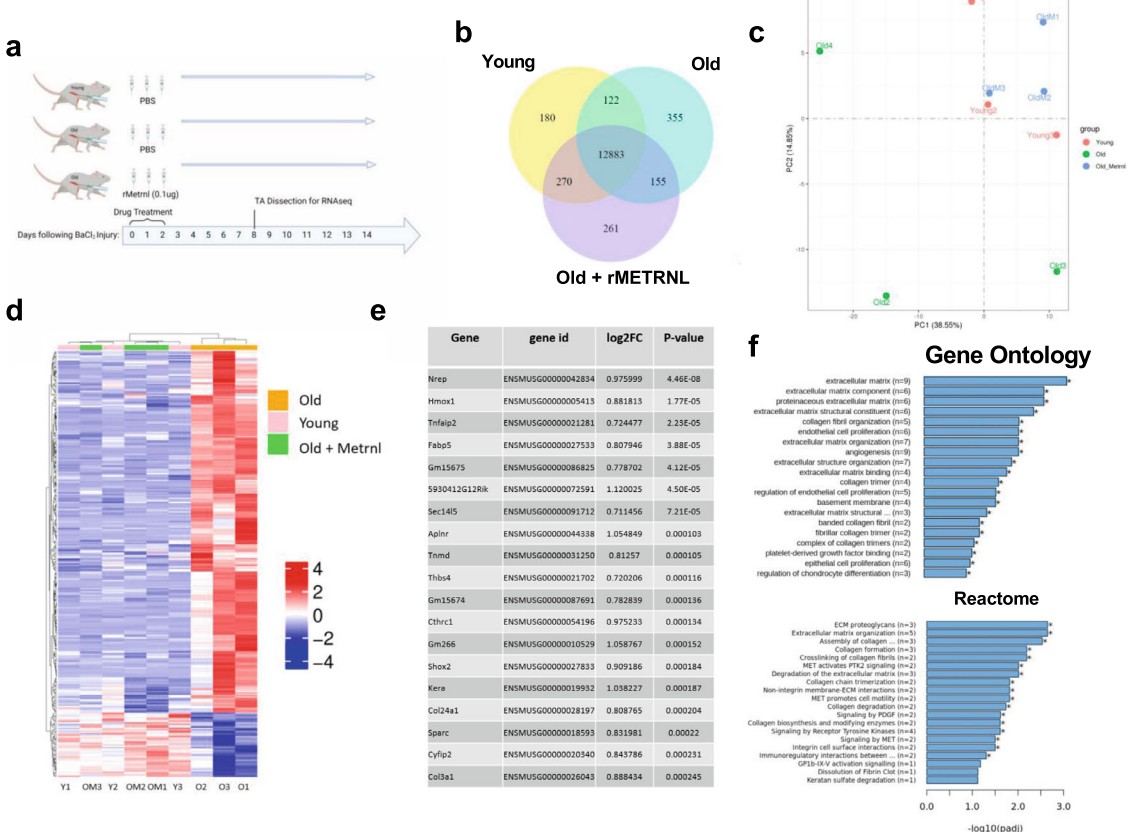

**Fig. 4 | rMetrnl rejuvenates the transcriptional profile of aged muscle regeneration. a** Graphical representation of experimental design (n = 3/group). **b** Venn Diagram of co-expressed transcripts between groups. **c** 2D Graphical representation of principal component analysis showing top two principal components of transcriptomic data. **d** Heatmap and dendrogram of transcript (y axis) expression level compared to mean expression across all groups (colormap) from each sample (x axis). **e** Top transcripts ranked by significance and log fold change when differential expression analysis was performed between old and old+rMetrnl groups; two-tailed t test was performed between old and old+rMetrnl and FDR adjusted to determine p value. **f** Gene set enrichment analysis performed to determine p values between old and old+rMetrnl groups specifically using Gene Ontology (Biological Processes) and Reactome databases. Graphics created with BioRender.com. Source data are available through GEO Accession ID GSE217037.

inferred pathways were IL1, PDGF, IGF, TNF among others (Fig. 5f). Further normalization and analysis of the subset of FAPs was performed and differential gene expression analysis allowed the categorization into specific FAP subpopulations, primarily differentiated by cell cycle behavior such as quiescent, actively dividing or an ECM-related gene signature (data not shown). There was a small clustering of cells that had a distinct gene signature from the other categories and was overrepresented by cells from the old PBS sample compared to young PBS or old rMETRNL samples (Fig. 5g, h). This subpopulation represents FAPs that have reduced signatures of cellular proliferation regulation, reduced peptidase activity, greater ECM secretion, and reduced cell cycle activity (Fig. 5i). Results from these scRNA-Seq inferential studies suggest an age-related immune cell phenotype early in regeneration, which is associated with a pro-fibrotic gene signature later in regeneration. Treatment with rMETRNL returns the aged

immune phenotype back to a youthful state, which is associated with the suppression of the pro-fibrotic phenotype.

## Fibro/adipogenic progenitor cells are responsive to post-injury rMETRNL in aged muscle

Next, we aimed to identify factors by which the rMETRNL-responsive macrophages signal to FAPs, leading to improved aged muscle regeneration. We performed fluorescence-activated cell sorting (FACS) on Sca1+, CD45−, CD11b−, CD31− cells (from here referred to as Sca1+ FAPs) from injured muscle-derived cell suspensions (Fig. 6a) to collect these FAP population for gene expression analysis. We measured higher levels of *Pdgfra*, *Col1a1*, and *Fn1* by RT-qPCR in the old PBS group compared to both the young PBS and old rMETRNL group (Fig. 6b). We also collected CD45+, CD11b+ myeloid cells (predominantly macrophages at this timepoint) and cultured these cells

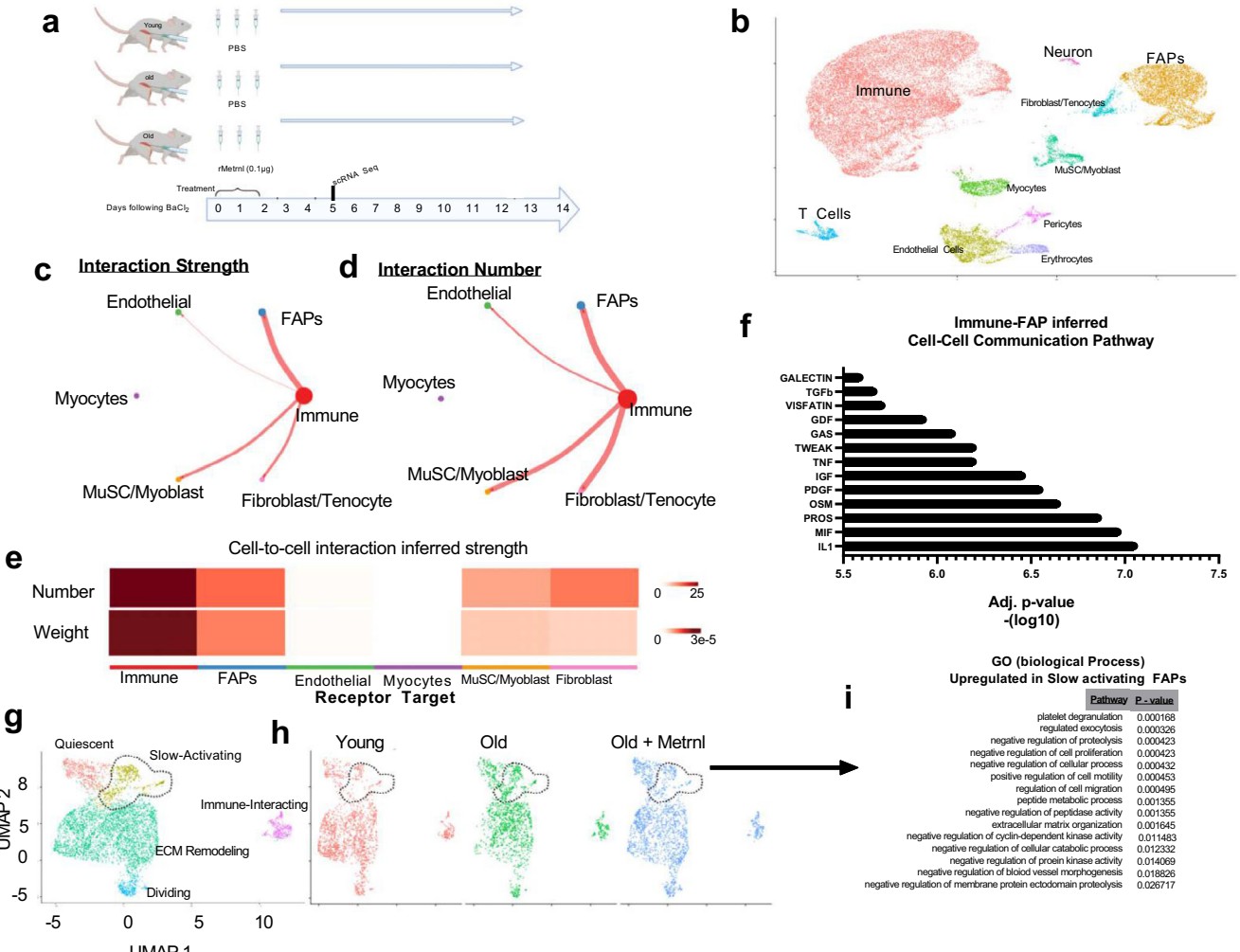

**Fig. 5 | Single-cell transcriptomics shows rejuvenation of aged muscle tissue by rMetrnl treatment 4 days post-injury. a** Graphical representation of the experimental design. **b** Combined UMAP reduction plotted in two dimensions with cell cluster identities labeled. **c**, **d** Cell-to-cell communication analysis assessing ligand/receptor pairings across cell types. Immune cells were chosen for ligand source and strength of interaction and the number of ligand/receptor pairs is indicated by chord thickness. Number of cells in a group is indicated by circle diameter. **e** Heatmap showing results from **c**, **d**. **f** Two-tailed *t* test was performed and FDR adjusted to infer cell-to-cell communication pathways upregulated with immune cell-expressed ligands and FAP-expressed receptors. **g** Subset analysis of FAP cells from **b** shown by UMAP reduction and separated by sample group (**h**). Dashed circle highlights distinct FAP population identified as overrepresented in old PBS cells. **i** Gene Set Enrichment Analysis was used to determine *p* value using GO (biological process) database on FAP cell subset circled from **g**, **h** when differential expression analysis was performed against all other FAP cells. Graphics created with BioRender.com. Source data available through GEO at Accession ID GSE217258.

for 36 h before analyzing levels of TNFα in the conditioned culture media. Results showed an overall decrease in TNFα concentration in the media conditioned by old PBS myeloid cells compared to both young PBS and old rMETRNL (Fig. 6c). Because TNFα is a known contributor to FAP apoptotic signaling[19], we analyzed Sca1+ FAPs at a later timepoint (7 days post injury) to determine if the TNFα contributed to altered proportions of FAPs in the muscle across the regenerative program. The Sca1+ FAPs in the old rMETRNL samples had a significantly higher frequency of the apoptotic marker AnnexinV (Fig. 6d). To rule out direct effects of rMETRNL on Sca1+ FAPs, we performed annexinV staining and metabolic flux analysis in vitro using FACS-sorted Sca1+ FAPs and saw no significant changes between the old and old + rMETRNL-treated Sca1+ FAPs (Fig. S5a–d). We also tested the direct effects of METRNL in vitro, on FACS-sorted muscle stem cell proliferation (Fig. S5e) and total satellite cell numbers at 3 days PI in vivo, after rMETRNL treatment (Fig. S5f). We failed to see any changes in proliferation or cell number with METRNL treatment.

## TNFα-induced FAP fibrogenic differentiation and apoptosis mediate the effects of rMETRNL in old muscle regeneration

These results led to the hypothesis that rMETRNL is contributing to greater amounts of secreted TNFα in old macrophages, which then signal directly to FAPs. We tested this hypothesis using an ex vivo media transfer system. We isolated and cultured macrophages and FAPs separately to allow treatment of macrophages with rMETRNL for conditioning media used in culturing FAPs. This allowed use of pharmacological inhibitors in the treatment of FAPs, directly, without impacting macrophage secretion as would occur in a co-culture approach. Young and old BMDMs were treated with rMETRNL for 36 h and conditioned media collected. This medium was then used to culture Sca1+ FAPs isolated from young and old uninjured muscle for 5 days before analyzing fibrogenic gene markers (Fig. 6e). We found significant upregulation in gene expression of *Pdgfra, Col1a1,* and *Fn1* in old FAPs cultured in old macrophage media compared to young FAPs cultured in young macrophage media. The age-related fibrogenic markers were reversed in old FAPs cultured in rMETRNL-treated

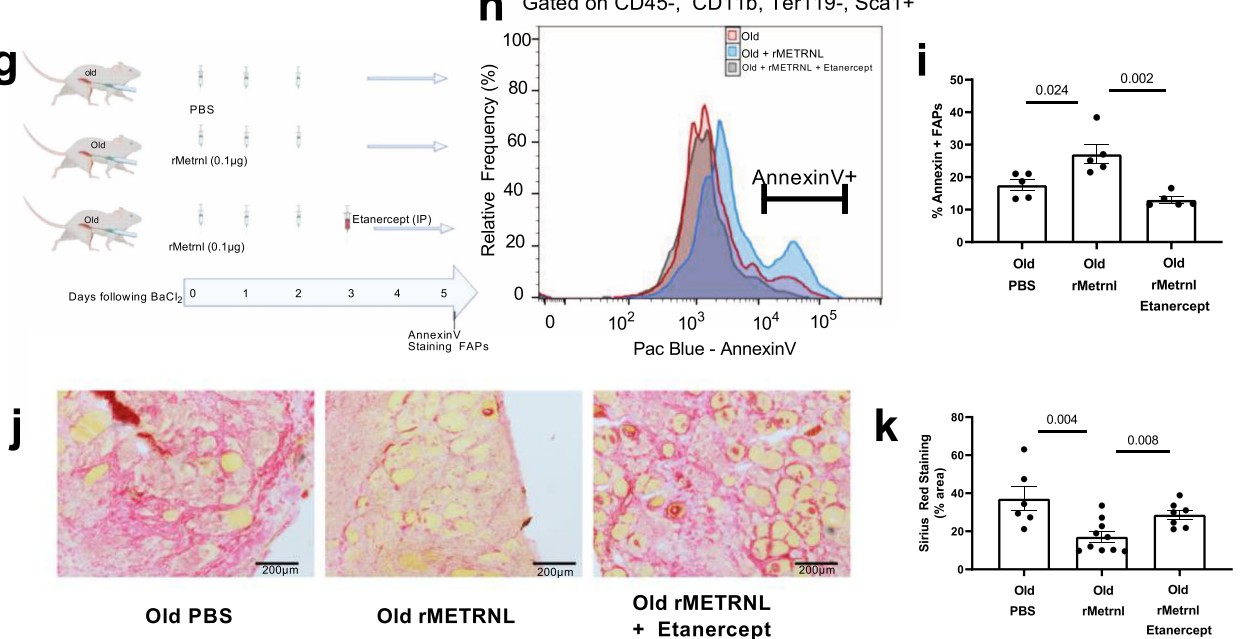

macrophage medium (Fig. 6f). This finding substantiated our previous in vivo data suggesting rMETRNL treatment altered muscle cytokine secretion, which affected FAP differentiation. To determine the necessity of TNFα signaling, old FAPs were cultured in rMETRNL-treated macrophage medium with 10 uM Etanercept (Embrel) to inhibit TNFα specific signaling. The inhibition of TNFα signaling negated the benefits of rMETRNL-treated macrophage media, promoting increased fibrogenic gene expression in the FAPs (Fig. 6f). These findings support our hypothesis that TNFα is acting to mediate the

immune-FAP signaling induced by rMETRNL. Knowing the limitations of this ex vivo approach (especially the atypical physiology of primary BMDM-differentiated macrophages), we attempted to translate these findings to an in vivo system. We performed a similar intramuscular rMETRNL treatment in old mice, as used in previous experiments, with a subset of mice receiving 1 mg/kg Etanercept via intraperitoneal (I.P.) injection following the final rMETRNL treatment (Fig. 6g). We hypothesized the rMETRNL treatment would stimulate FAP apoptosis measured by AnnexinV and that this could be reversed by inhibition of

**Fig. 6 | Metrnl rejuvenates aged muscle regeneration through macrophage-derived TNF-induced FAP apoptosis. a** Graphical representation of the experimental design for **b**–**d**. **b** FACS-sorted cells Sca1+ FAPs were collected and mRNA analyzed by RT-qPCR. Genes are shown relative to 18 s which remained unchanged between biological groups ($n = 6$ animals/group); one-way ANOVA was used with Fisher's LSD post hoc analysis. **c** FACS-sorted CD45+, CD11b+ cells were collected from 3 days PI mice and cultured for 36 h to allow cytokine secretion into culture media ($n = 3$ animals/group). Culture media was then subjected to ELISA analysis for TNF concentration; one-way ANOVA was used with Fisher's LSD post hoc analysis. **d** Sca1+ cells from 7 days post injury were collected by and stained with AnnexinV and analyzed by flow cytometry to quantify apoptotic cell percentage ($n = 3$/group); one-way ANOVA was used with Fisher's LSD post hoc analysis. **e** Schematic representation of ex vivo culture system using conditioned media collected from BMDM-differentiated M0 macrophages from young and old mice. This conditioned media was then used to culture Sca1+ FAPs isolated by FACS from young and old

muscle. This allowed the separate treatment of M0 macrophages with rMETRL and treatment of FAPs with Etanercept. **f** mRNA from **e** analyzed by RT-qPCR ($n = 6$/group); one-way ANOVA was used with Fisher's LSD post hoc analysis. **g** schematic representation for in vivo rMETRNL treatment combined with TNF inhibitor experiment. **h** Representative frequency distribution and **i** summary data for flow cytometry of AnnexinV gated on Sca1+ FAPs at 5 days post BaCl₂ injury; ($n = 5$ mice/group); one-way ANOVA was used with Fisher's LSD post hoc analysis. **j** Representative micrographs and **k** summary analysis of Sirius Red staining of collagen from tibialis anterior muscles 5 days follow BaCl₂ injury ($n = 6$ old, 10 rMetrnl, 7 Etanercept muscle sections analyzed representing five animals/group); one-way ANOVA was used with Fisher's LSD post hoc analysis. $p$ value is indicated with connecting line between comparison groups. Source Data are provided as a source data file. Data are presented as mean values ± SEM. Graphics created with BioRender.com.

TNF using Etanercept in vivo. Our results showed that rMETRNL administration early during regeneration (Day 0–2) stimulates clearance of FAPs measured by a ~40–60% increase annexinV labeling ($p = 0.024$) which could be abrogated by the systemic TNFα inhibitor Etanercept ($p = 0.002$, Fig. 6g–i). To determine if METRNL-induced effects on FAPs would translate into differences in muscle fibrosis, we performed histological analysis on aged muscle sections, post injury, treated with rMENTRL with or without etanercept. When we analyzed the percent area covered by stained collagen, the results showed high fibrosis in the Old PBS control muscle, which was reversed with rMETRNL treatment (F6J-K). In contrast, co-treatment with etanercept negated the benefits of rMETRNL to reduce muscle fibrosis (F6J-K). These results support the role of METNRL is regulating age-related fibrosis, in a TNFα-dependent manner.

## Discussion

The immune response during muscle repair has emerged as a critical process for successful regeneration and is, in part, regulated by the macrophage-secreted factor Meteorin-like (METRNL)[5]. Considering the importance of METRNL during the regenerative process, this presents a natural therapeutic target for improving regeneration in aged muscle. In this study, we demonstrate that incomplete muscle regeneration in aging is associated with insufficient production of METRNL, and this can be counteracted by exogenous delivery of METRNL to the damaged tissue. We extend these observations by showing the therapeutic efficacy of an injectable recombinant METRNL therapy to enhance aged muscle regeneration, evidenced by cellular, transcriptomic, functional, and histological improvements. We then detail a mechanism by which METRNL treatment restores aged macrophage TNFα secretion, contributing to fibro/adipogenic progenitor apoptosis and suppression of pro-fibrotic differentiation and fibrosis.

The ability of *Metrnl* to reprogram aged macrophages suggests that METRNL may act as a therapeutic in conditions where the pro-regenerative immune response is insufficient. In our initial time course experiment, we found a dramatic reduction of *Metrnl* following muscle damage in aged mice. We were able to recapitulate the age-related reduction in *Metrnl* expression in cultured BMDMs. These data are in agreement with our previously published proteomics of secreted proteins from young and old macrophages, which revealed METRNL as one of the top downregulated secreted proteins from old macrophages[20]. Our blood exchange models (parabiosis and bone marrow transplantation) support the rejuvenating effects of a youthful immune system to return Metrnl expression after injury. However, we also highlight the necessity of *Metrnl* to achieve full effect.

Age-associated detriments to muscle repair result in incomplete restoration of muscle contractile capacity and greater non-contractile tissue and fibrosis[21]. Evidence by Kuswanto et al.[17] suggests detriments to innate immune signals contribute to age-related degeneration in

aged mice. This suggests that a lack of METRNL may be a contributing factor to the age-related macrophage alterations seen during muscle regeneration. While several other macrophage-derived peptides have been shown to be required for muscle repair, (i.e., IL-10[22], CCL2[23], IGF-1[24], ADAMTS1[25]), few have been tested in the context of aged muscle regeneration. Although we used several models of exogenous cell-based delivery of METRNL, we also showed that direct intramuscular delivery of the recombinant peptide was sufficient to enhance aged muscle regeneration and functional capacity through its modulation of macrophages in agreement with findings by Ushach et al.[11]. We next sought to outline the connection between altered macrophage phenotype induction by rMETRNL and enhanced regeneration.

Using bulk and single-cell transcriptomic approaches, we inferred cell-to-cell communication signals originating in immune cell sub-populations targeting FAPs as the effector cell population. Since their identification in 2010[26], FAPs have been revealed as key mediators of homeostasis and repair in several areas of muscle physiology. In healthy muscle regeneration, the number of intramuscular FAPs transiently expand to 2–4 times greater than that of uninjured control muscle before returning to baseline around 9 days post injury[19]. In pathological conditions, uncontrolled FAP clearance can lead to unsuccessful muscle regeneration and fibrosis. This has been studied in murine models of Duchene's Muscular dystrophy, where FAPs play a contributing role to the progressive muscle fibrosis associated with the diseases' degenerative progression[27]. Furthermore, Lemos et al[19]. helped to outline a relationship between FAP differentiation, TNFα and FAP apoptosis—a clear cell-to-cell signaling network altered in aged muscle according to our data. Here, we have provided novel data on age-associated detriments to FAP population expansion and clearance in response to BaCl₂-induced damage. We have also demonstrated their propensity for reduced apoptosis post injury, a time when healthy muscle has highly upregulated FAP clearance. These data outline potential aspects of FAP biology, which could be targeted by therapeutic strategies to improve muscle homeostasis and repair in aged muscle.

To our knowledge, no therapies have been tested which aim to manipulate FAP responses in aged muscle to improve recovery following damage. We present data demonstrating *Metrnl*-mediated enhancements in muscle regeneration in multiple mouse models. We then demonstrate the rejuvenating properties of METRNL are mediated by the effects of TNFα on FAP fibrogenesis and apoptosis in aged muscle. Currently, the molecular cross-talk between immune cells and FAPs is limited, especially in the context of aging. However, our data corroborate those of Lemos et al, who showed the macrophage-derived cytokine TNFα plays an integral part in FAP apoptosis and clearance in the *mdx* mouse[19]. These findings demonstrate a specific mechanism for rMETRNL treatment to enhance muscle regeneration, but they also show a proof-of-concept that FAPs provide a therapeutic cellular target to combat age-related loss in muscle regenerative capacity. Using a clinically relevant approach, we demonstrate the

rejuvenating effects of *Metrnl* can be attained by intramuscular administration of the recombinant peptide, at therapeutic doses, in the short-term following muscle damage.

Together, these data demonstrate a novel age-related decline in *Metrnl* expression following muscle damage contributing, in part, to incomplete muscle regeneration. We have outlined cellular strategies to restore *Metrnl* in mouse models of physiological and BaCl$_2$-induced muscle damage through adenoviral transduction, shared circulation, and chimera generation. We then demonstrated the efficacy of intramuscular administration of an rMETRNL as a rejuvenating biologic to enhance aged muscle regeneration. Lastly, we outlined a cellular mechanism for rMETRNL action through macrophage phenotype and secretion during the initial stages of regeneration, leading to a downstream molecular mechanism of TNFα-induced FAP modulation and apoptosis.

## Methods

### Animals
All animal experiments and care followed the guidelines and were approved by the Institutional Animal Care and Use Committee at Duke University Medical Center. For all experiments, C57/BL6J mice were acquired through the National Institute of Aging animal colony and used for experiments at 3–4 months of age (labeled as 'Young') or at age 24–26 months ('Old'). For parabiosis and bone marrow transplant experiments, Metrnl knockout mice (Metrnl$^{-/-}$) were previously described[5]. All mice were house in climate-controlled facility with temperature maintained at $22\,°C \pm 1\,°C$, humidity maintained between 30–70%, and light/dark cycle of 12 h/12 h. One experimental endpoints were reached, mice were euthanized by CO$_2$ asphyxiation for >5 min followed by decapitation as secondary confirmation.

### AAV infection
Where indicated, C57/Bl6J, male mice were transduced with 100 µl AAV8-null or AAV-Metrnl vectors at $3.0 \times 10^{11}$ vg/mouse administered by tail vein injection 14 days prior to subsequent experimentation.

### Muscle injury
Barium chloride (BaCl$_2$) was used to induce muscle damage in tibialis anterior muscle via intramuscular injection of 30 uL of 1.2%(vol/vol) BaCl$_2$ in PBS using 30 g needle[28]. Uninjured controls were injected with an equal volume of PBS. For experiments shown in Fig. 1d–g, muscle damage was induced by exhaustive downhill treadmill running, 22 m/min at a −15 degrees angle[29] with average time to exhaustion of 120 m.

### Parabiosis
Anastomosis surgery was carried out on female mouse pairs 3 weeks prior to muscle injury experiments[5,20,30]. Under isoflurane anesthesia, mice were randomly selected for pairing with extreme differences in size precluding some pairings. Skin and fascia were cut from elbow to knee before suturing to close the wound. This established isochronic pairs (old to old), heterochronic wild-type pairs (young WT to old WT), and heterochronic Metrnl$^{-/-}$ pairs (young Metrnl$^{-/-}$ to Old WT).

### Reconstituted bone marrow chimera
Male Recipient mice were irradiated with 900 cGy and injected via tail vein injection with donor bone marrow cells ($1 \times 10^6$ cells in 200 µL PBS). Engrafted cells were allowed to establish for 2 months before muscle injury experiments and verified by flow cytometry[5,20].

### Real-time qPCR
RNA was extracted from whole muscle using Trizol reagent and glass bead homogenizer before purification using Qiagen (Germantown, MD, USA) spin column. Reverse transcription was performed using SuperScript VILO cDNA synthesis kit (Life Technologies, Waltham, MA, USA). Real time PCR was performed on Quant Studio 6 system (Life Technologies) using Power SYBR Green PCR mastermix (Life Technologies) and appropriate primers listed in supplementary table 1.

### Bone marrow-derived macrophage culture
Primary mouse bone marrow-derived macrophages (BMDMs) were obtained by flushing mouse tibia and femurs with DMEM using a 27-guage sterile needle into a collection tube. The marrow was dispersed by passing the sample through the same needle multiple times. Red blood cells were lysed using ammonium chloride and remaining cells were seeded on 10 cm tissue culture plates in DMEM supplemented with 10%(v/v) fetal bovine serum mixed with 30%(v/v) L929 cell-conditioned medium for 5 days at 37 C, 5% CO$_2$. RNA from these cells was then collected for RNA sequencing analysis. Conditioned media was collected for proteomics analysis of secreted proteins. To activate the BMDM, they were left untreated for 2 days (Control), treated with 10 ng/mL of IL-4, or treated with 50 ng/mL Salmonella LPS for 2 days RNA was collected and analyzed by qPCR, conditioned media were collected for ELISA of METRNL.

### Muscle histology
Muscle architecture was analyzed by freezing tibialis anterior muscle embedded OCT by dipping in liquid nitrogen-cooled isopentane. 10 uM cross sections were cut using a cryotome and stained using hematoxylin and eosin for experiments in Fig. 3 and picosirius red for Fig. 6. Images were taken using Olympus microscope and analyzed using Olympus CellSense software to determine area of individual fibers. CSA was calculated by determining area of >100 fibers across three images for each sample and using this result to average with each sample from that experimental group. For picosirius red staining, all images were taken using identical camera and light settings. RGB images were imported to ImageJ, thresholds were set to cover Red stained regions and percent of threshold pixels was determined. All analysis steps were used to generate a macro script which was applied to all images from the experiment to maintain consistency.

### In situ force measurement
Tibialis anterior (TA) muscle contractile properties were measured using the Aurora Scientific 1300-A whole mouse muscle test system. Briefly, the mouse was anesthetized using 2.5% isoflurane and body core temperature maintained at 37 °C using a heated platform. The hind limb was shaved and the TA muscle was surgically exposed and the distal end of the TA tendon sutured to the force transducer. To induce muscle contraction, the sciatic nerve was exposed by an incision along the thigh and the peroneal branch identified. Needle electrodes were placed on the peroneal branch of the sciatic nerve to specifically induce contraction to the TA. Throughout the experiment, saline was applied to the exposed TA muscle to ensure the muscle remained fully hydrated. To assess muscle function the optimal length (L0) and current amplitude were determined for each individual mouse. A force-frequency relationship was measured by applying 0.25 s trains of 1, 10, 20, 50, 75, 100, 120, and 150 Hz with a 60 s rest period between each elicited contraction. Data were recorded and analyzed using the Lab-View-based DMC and DMA software (Version 3.12, Aurora Scientific, Aurora, ON, Canada). At the end of each experiment, the distal TA tendon was removed and TA muscle weight was measured. The muscle CSA was determined based on a muscle density of 1.06 g/cm$^3$ and fiber length to Lo ratio of 0.6 as described previously[31]. All force measurements and tissue collections were performed blinded to experimental group identity.

### RNA sequencing
RNAseq was performed by Novogene Co. (Durham, NC). Total RNA from muscle was extracted as detailed above. 1 µg of RNA was used to generate cDNA libraries using NEBNext Ultra RNA Library Prep Kit

(NEB, USA) per manufacturer's instructions. Libraries were sequenced on NovaSeq 6000 platform (Illumina) and paired-end reads were generated and mapped to mouse genome. Quantification and differential gene expression were performed using DESeq2 R package follow by GSEA[32].

## 10x genomics-based single-cell analysis

Single-cell RNAseq was performed on single-cell suspensions collected from pooled triplicate samples from tibialis anterior muscles of young PBS-treated, old PBS-treated, and old-rMETRNL-treated mice 96 h following $BaCl_2$ induced muscle damage. 15,000 cells from each of the three samples were used to generate cDNA. On 10X Genomics Chromium Drop-seq platform, cells were resuspended in a mastermix containing reverse transcription reagents, ×10 barcode, transcript identifiers and lysed in nanoliter scale gel bead emulsions. When each cell/bead was lysed, reverse transcription began on all transcripts in a single bead emulsion and identifiable by ×10 barcode. cDNAs were cleaned, quantified, and sheared for library construction by end-repair and A-tailing. Final libraries contained bridge amplification primers. Paired-end sequencing was performed on 45,000 cells per library on Illumina 2500 sequencing platform at 50,000 reads/cell. Single-cell expression data was performed in 10X Genomics Cell Ranger software v1.3 to demultiplex raw FASTQ reads and align with mouse genome. Seurat package[33] was used to perform quality control analysis, sample scaling, and normalization. Cell clusters were identified using Seurat FindClusters function and visualized using UMAP dimensionality reduction. Cluster identification was performed using standard gene markers for positive and negative selection. Further subset analysis of fibro/adipogenic progenitor cells (FAPs) was performed using Seurat Subset function, and rescaling/normalizing cell subset. CellChat package[18] was used for cell-to-cell communication pathway inference. For in silico analysis performed in Supplemental Fig. 1d–g, processed and integrated data were generously shared by B. Cosgrove associated with the publication McKellar et al.[15] and analyzed using similar packages as stated above. All functions were used from listed packages without change and scripts are available upon request.

## Flow cytometry and cell sorting

Mouse muscle was collected, digested, and prepared into a single-cell suspension for flow cytometry or FACS using the methods detailed by Liu et al.[34]. Triceps, quadriceps, hamstrings, tibialis anterior, gastrocnemius, soleus, and plantaris muscles were collected from uninjured mice, or tibialis anterior muscles were used in mouse injury experiments. Muscles were dissected, minced using scissors, digested in 10 mL of F10 + 10%(v/v) horse serum and 1000 u/mL Collagenase, Type 2 (Worthington) for 60 min at 37 °C in a shaking water bath, washed in F10 media, digested again for 30 min at 37 C in F10 + 10%(v/v) horse serum + 1.1 U/mL dispase (Gibco) and 100 U/mL type 2 collagenase, washed, passed through 40 uM cell strainer and stained with antibodies for flow cytometry. Cells were identified on flow cytometers by forward scatter and side scatter parameters. For analysis of leukocytes, negative selection markers were AF488-Ly6G and AF488-SiglecF, and positive selection markers CD45-APC and CD11b-PE-Cy7, F4/80-PE with Ly6C-PB hi and low determined by gating on unstained and fluorescence minus one (FMO) control samples (shown in Fig. 3b and Fig. S2e). For analysis of FAP cells, cells were negative gated for APC-CD45, APC-CD11b, APC-Ter119, and APC-CD31, and PE-VCAM1 populations before selecting SCA1 + cells (gating and FMO controls shown in Fig. S4h, i). For AnnexinV analysis, these Sca1+ cells were positively gated before analysis of PB-AnnexinV. Flow cytometry and FACS were performed on Sony MA900 cell sorter equipped with four lasers and appropriate detector filter sets. Cell sorting was performed in biosafety cabinet at 4 C using 100 μM microfluidic chip at a pressure near 20PSI. Cells were collected in media used for culturing but washed once prior to seeding to remove FACS sorting sheath fluid.

## Fibro/adipogenic progenitor culture

FAPs were collected by FACS as described above and cultured at 37 C, 5% $CO_2$ humidified incubator in F10 + 10%(v/v) Horse serum and 10 ng/mL bFGF. Where indicated, FAP culture media was mixed 50/50(v/v) with macrophage-conditioned media.

## Chemical reagents

Etanercept (Sigma, Y0001969) was used at 100 uM in vitro and 1 mg/kg in vivo delivery via intraperitoneal injection in 100 uL saline solution. bFGF (Gibco, #13256-029) was used at 10 ng/mL, rMETRNL (R&D Biosystems) was used at 100 ng/mL in vitro and 0.1 ng/dose in vivo. Antibodies used include the following: APC anti-mouse CD45 (1:200, Biolegend 157605), APC anti-mouse CD11b (1:200, Biolegend, 101211), PE anti-mouse CD106 (1:50, VCAM, Biolegend 105713), FITC anti-mouse Ly6A/E (SCA1, 1:50, Biolegend, 108105), Pacific Blue anti-mouse Ly6C (1:200, Biolegend, 128013), AlexaFluoro 488 anti-mouse CD170 (SiglecF, 1:200, Biolegend 155524), AlexaFluoro 488 anti-mouse Ly6G (1:200, Biolegend 127626), APC anti-mouse F4/80 (1:200, Biolegend 123116), Pacific Blue AnnexinV (Biolegend 640926), PE-Cy7 anti-mouse CD11b (1:200, Biolegend 101216). Mouse Meteorin-like DuoSet ELISA (R&D Systems DY6679).

## Statistical analysis

Statistical analyses performed included Student's $t$ test (unpaired, two-tailed) between pre-planned comparisons of two groups, one-way and two-by-two analysis of variance tests were performed using Fisher's LSD post hoc analysis when significant interactions were seen. All bar graphs presented are means ± SEM and $\alpha$ was set at $p < 0.05$. The number of replicates for each experiment is indicated in respective figure legends as well as defined markers for significant comparisons or $p$ values.

## Reporting summary

Further information on research design is available in the Nature Portfolio Reporting Summary linked to this article.

## Data availability

Single-Cell RNA sequencing data (Fig. 5) has been deposited to GEO at Accession ID GSE217258. Bulk RNA sequencing associated with Fig. 4 is available at GEO Accession ID GSE217037. Single Cell RNA sequencing associated with Supplementary fig. 1 was previously published[15] and is available at Accession ID GSE143437. Supplementary data 1 includes RNA sequencing results related to Supplemental Fig. 1a. Source data are provided in this paper.

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

## Acknowledgements

The authors thank Duke's Cancer Center Flow Cytometry Core and Duke's Molecular Genomics Cores for support on this project. The authors acknowledge Sophia DeLuca for assistance with animal protocols related to in situ force measurements. J.P.W. was supported by funds from Duke Aging Center/Pepper Center P30-AG028716, Borden Scholar Award through Duke University, NIH/NIA grant K01AG056664, and R21AG065943. D.E.L. was supported by NIH training grant T32HL007057. L.K.M. was supported by NIH/NIDCR grant K08DE031029. G.S.B. was supported by NIH/NIA grant R21AG067245. GAT was supported by NIH/NIAID grants AI145929 and AI148243. N.B. supported by NIH grant R01AR070543.

## Author contributions

D.E.L., L.K.M., A.B., G.S.B., and J.P.W. Conceived and/or designed experiments. D.E.L., L.K.M., A.B., G.S.B., Y.L., A.K., N.B., and J.P.W. Acquired, analyzed, and/or interpreted data. D.E.L. Drafted the manuscript. D.E.L., L.K.M., A.B., G.A.T., G.S.B., and J.P.W. Revised the manuscript. All authors approved the final version. J.P.W. Supervised the project.

## Competing interests

The authors declare no competing interests.
