## [Peer Review File · Nature Communications]

Meteorin-like is an injectable peptide that can enhance regeneration in aged muscle through immune-driven fibro/adipogenic progenitor signalingREVIEWER COMMENTS

Reviewer #1 (Remarks to the Author):

This is an interesting paper that shows the effect of meteorin-like on myogenic regeneration in young vs old mice, suggesting that it is reduced in aging and that replenishing exogenously may restore some of the regenerative capacity of the tissue. In general, the basic observation seems solid and the finding would be of interest to a wide readership. However, the analysis of the underlying effects is underdeveloped. In addition, issues of experimental design and data presentation make some of the results hard to fully understand.

The presentation of figure 3 b and d is confusing... what exactly is represented on the Y axis? If I understand correctly this represents CD45 and CD11b on the same fluorescent channel, which is inappropriate if the authors want to draw conclusions about macrophages. Why are the FACS plots cut off on the Y axis? I could not find a detailed description of the gating in either the text, figure legend or methods. This should be shown inclusive of FMO controls used to set the gates for quantification. How were neutrophils, eosinophils etc excluded from this analysis? Which Ly6C antibody was used and does it crossreact with Ly6G?

There is great variability in the exact timing of macrophage switching during muscle regeneration. This analysis should be carried out as a time course to assess whether the differences observed at the early time represent a change in the timing. This seems likely as the plots in D, at 7 days, do not show much difference. Is any of the differences reported in the associated plots statistically significant? Is there an effect on the absolute number of infiltrating myelomonocytic cells? Is there an effect on infiltrating lymphocytes?

In figure 4, the three old samples are at opposite corners of the PCA plot, indicating high variability... please discuss. The methods should contain more details on the workflow used to generate the data in this figure. For example is the dendrogram/heatmap in 4d generated using all genes, or a subset? Could not find this info.

In figure 6 a few plots containing quantification of flow cytometry results are presented. The plots used for quantification should be shown with appropriate controls, ideally in the supplementary.

The results reported in fig 6H are very interesting, but poorly representative of the in vivo situation. BMDM are not equivalent to those found in skeletal muscle, and while these experiments suggest a potential mechanism, this should be tested in vivo. Can etanercept block the effects of MTRNL in old animals? This should be easy to provide and it is absolutely critical.

Reviewer #2 (Remarks to the Author):

The present work is focused on the role of Meteorin-like (Metrnl) in muscle regeneration during ageing. The authors convincingly showed that Metrnl decreased in aged muscle after injuries. By transfecting muscles with AAV-virus expressing Metrnl it was sufficient to restore grip strength and time to fatigue after eccentric contraction damage in aged mice. Then the authors used parabiosis, bone marrow transplant and Metrnl injection to further prove the role of Metrnl in muscle rejuvenation. The ability of Metrnl to rescue muscle regeneration was also confirmed by using Metrnl knockout mice, which fails to rescue the regenerative capacity of old mice in parabiosis or bone marrow transplant experiments. To identify the insights that justify Metrnl action, the authors performed scRNA seq and found changes in a subpopulation of FAPs, which become more fibrogenic in old mice but not in Metrnl-treated aged animals. Finally they identify TNFa as the culprit for the persistence of the pro-fibrogenic FAPs in aged mice. In fact, Metrnl restores TNFa expression in macrophages causing apoptotic cell death of FAPs and therefore reducing the pro-fibrogenic program. I like the elegant experimental design that the authors used to prove the role of Metrnl in muscle regeneration during ageing. The data are convincing and sustain this correlation. However, the drawback is that the insights that explain such beneficial effect are weak. The authors should consider the following points.

Point1. The authors claimed that macrophages are the major source of Metrnl during injury. However, this should be proven. It is strange that the peak of Metrnl expression happens ad day 1 after BaCl2 injection, a time in which the most abundant cells are polymorphonuclear neutrophils and not monocytes/macrophages. Please use scRNAseq or multiple FISH to prove that after eccentric contraction or BaCl2 injection, macrophages are the only cell expressing Metrnl.

Point2. The Metrnl effect on CSA mst be better characterized. Is this due to increase myoblast proliferation and fusion or to an enhancement of protein synthesis in regenerating myofibers? Both the options are sustained by the presence of IGF1 and TGFb cytokines among the top differentially expressed genes in Fig 5F and both are well known modulators of protein synthesis and myogenic differentiation.

Point3. Grip strength is a modest indicator of muscle function because it can be affected by pain and other inputs. Please measure muscle force by ex-vivo or in vivo experiments. Muscle force should be also tested in Fig3 experiments.

Point4. The connection with FAPs and TNFa expression is weak. In case that impairment of TNFa would affect FAPs survival, why young and old animals show the same level of Sca1+ cells (Fig 6e) despite the young mice show higher level of TNFa secretion (Fig 6d)? If this idea is true they should mirror the old+ Metrnl mice. Indeed, they show an increased cell death (Fig6f) compared to old mice.

Point5. How TNFa is modulated by Metrnl? Which pathway is involved?

Point6. Fig 7g-h suggest a potential role of TNFa pathway in regulation of pro-fibrogenic FAPs but does not show that it is the culprit of the muscle phenotype in aged mice. To sustain the authors hypothesis, authors must prove in vivo that: 1) the profibrogenic FAPs are indeed responsible for the decreased CSA and muscle force in BaCl₂ treated mice and after eccentric contraction, 2) the TNFa expression of macrophages affect the FAPs subpopulation, the decreased CSA and muscle force in BaCl₂ treated mice and after eccentric contraction.

My Co-authors and I would like to thank each of the reviewers for the time and attention to detail that is evident by the reviews. We were excited to see from the reviewers' introductory comments that our manuscript was received with enthusiasm as well as their clear understanding of the story which we tried to present. The clear and accurate summaries given by the reviewers gave us confidence in their understanding of our work which allowed us to focus our attention on the comments and suggestions brought forth by their expertise.

In lieu of the points brought out by the reviewers, my co-authors and I are happy to include several major revisions to the original manuscript. We are excited that these new experiments bolster our original findings and give new insight into the molecular mechanism of Meteorin-like in muscle regeneration and aging. Specifically, we performed and now include data from these additional studies:

- scRNA-seq analysis of *Metrnl* gene expression using a comprehensive muscle regeneration dataset made recently available (Sup Fig 1).
- A time course of flow cytometry analysis of macrophage subtypes using a completely revised cell labeling and gating strategy (Fig 3, Sup. Fig 3).
- In Situ functional analysis of rejuvenated muscle regeneration measured by force generation in rMETRNL-treated old mice (Fig 3)
- In vivo TNF α inhibition during muscle regeneration in rMETRNL-treated old mice (Fig 6).

We are excited to include these new data in our revised manuscript and are hopeful that we were able to address the concerns brought forth during the initial review. Details regarding these revisions in the context of the reviewer comments is given below with the original comments copied verbatim in red font.

REVIEWER COMMENTS

Reviewer #1 (Remarks to the Author):

This is an interesting paper that shows the effect of meteorin-like on myogenic regeneration in young vs old mice, suggesting that it is reduced in aging and that replenishing exogenously may restore some of the regenerative capacity of the tissue. In general, the basic observation seems solid and the finding would be of interest to a wide readership. However, the analysis of the underlying effects is underdeveloped. In addition, issues of experimental design and data presentation make some of the results hard to fully understand.

The presentation of figure 3 b and d is confusing... what exactly is represented on the Y axis? If I understand correctly this represents CD45 and CD11b on the same fluorescent channel, which is inappropriate if the authors want to draw conclusions about macrophages.

We appreciate the concern by the reviewer on the questions about both labelling and gating in our flow cytometry analysis. In the original submission, this was unintentionally omitted. In our original analyses, our flow cytometry panel did (as the reviewer points out) include multiple labels on a single fluorescence channel to accommodate the capacity of our flow cytometer. In lieu of the concerns raised by the reviewer regarding our flow cytometry experiments, my co-authors and I have completely reworked our approach to these macrophage phenotyping experiments in several key aspects:

- 1) We have repeated the animal experiments that were used for the macrophage phenotyping (Figure 3A-E) with a revised flow cytometry strategy. Importantly, data from these new experiments strongly agrees with data from our previous experiments while showing reduced variability and offering greater confidence thanks to the reviewer suggestion.
- 2) We have a revised labeling and gating strategy which we now employ to allow more clear and interpretable data regarding macrophage subpopulations among other immune subtypes.
- 3) We have clarified our new label and gating strategy by including an example population ancestry in Figure 3B with supplementary cell population data shown in Supplemental figure 3a, b and c and fluorescence-minus-one control examples used for gating and compensation shown in supplemental figure 3d.

Why are the FACS plots cut off on the Y axis?

Our previous FACS plots were cut off on the lower side of the Y axis (previously APC) because the ancestry used to generate the plot (which we had not then presented) included a parent population that was gated to include APC high features. Our revised experiments now shown in Figure 3C and 3E display similar results but hopefully with more clarity. At our chosen time points (3 days and 5 days-post BaCl₂ injection), there are very few cells labelled as CD45+, CD11b+, F4/80 low suggesting minimal lingering monocytes this long following injury. As such, the representative 2D dot plots that we show (Pacific blue - Ly6C vs. PE – F4/80) have been gated to exclude monocytes and only show F4/80 high populations. We hope this has been clarified by our revised labeling strategy and inclusion of sample ancestry in Figure 3B.

I could not find a detailed description of the gating in either the text, figure legend of methods. This should be shown inclusive of FMO controls used to set the gates for quantification.

The reviewer raised legitimate concerns regarding the labelling and gating strategy of flow cytometry experiments. We have attempted to clarify our presentation and approach by including the example gating strategy with ancestry in Figure 3B. Additionally, we have also added sample histograms of each fluorescent channel utilized in our experiment displaying a fluorescence-minus-one control with a sample labeled by all colors (Supplemental figure 3D).

How were neutrophils, eosinophils etc excluded from this analysis? Which Ly6C antibody was used and does it crossreact with Ly6G?

The reviewer raises an important question regarding the exclusion of various myeloid cell types in our flow cytometry analysis. As mentioned, these concerns over our previous experiment led us to redesign our labeling and gating strategy. Our revised experiments have included Ly-6G (targeting neutrophils) and Siglec-F (targeting Eosinophils); however, it should be highlighted that these two cell surface markers used the same fluorescence channel (FITC) in our experiments. We do not believe this alters the interpretability of our data because Ly-6G and Siglec-F were both excluded by our gating strategy (we gated on FITC-low excluding neutrophils and/or eosinophils). We have tried to make this clear by including the cell surface labels in our gating/ancestry sample (Figure 3B), by including this text to the figure legend and methods sections of the text, and by highlighting the combined population statistics in supplementary figure 3A,B and C.

There is great variability in the exact timing of macrophage switching during muscle regeneration. This analysis should be carried out as a time course to assess whether the differences observed at the early time represent a change in the timing. This seems likely as the plots in D, at 7 days, do not show much difference.

In our initial experiments, we did see a clear delineation among macrophage types between young and old groups at the 3 day time point that was fully transitioned by day 7. However, based on the reviewer comment, our new figure has been edited to include a timecourse including initial (1day), intermediary (3 day) and late (5day) timepoints following injury. We see no differences in F4/80+ macrophage subtypes due to rMETRNL at the 1 day timepoint but we still identify statistically significant difference between macrophage populations at the 3 day timepoint which are gone by the 5 day timepoint signifying a near completion of this portion of the inflammatory response. We hope that these revised experiments – which now include a time course of macrophage phenotyping for all three groups at three key timepoints -- alleviates the concerns raised by the reviewer regarding the dynamics of macrophage timing.

Is any of the differences reported in the associated plots statistically significant?

The reviewer points out a clear omission on our part where we failed to include statistical comparisons within our data presentation. We have now included the statistical notation within the graphs. To enhance clarity, we have also separated into two graphs the Ly-6C low and high quantification which we previously overlaid (now, Figure 3c, d, e graphs).

Is there an effect on the absolute number of infiltrating myelomonocytic cells? Is there an effect on infiltrating lymphocytes?

The reviewer raises an interesting question regarding the overall number of infiltrating immune cells into the muscle syncytium in response from rMETRNL. These data were readily quantified by our flow cytometry analyses and we now include the results in Supplemental figure 3A-C. The results indicate minimal statistical differences among

leukocytes (CD45+), myeloid lineage (CD45+ AND CD11b+), or our combined fluorescence for either Neutrophils (CD45+, CD11b+, Ly-6G+) OR eosinophils (CD45+, CD11b+, SiglecF+). This was true for both the 3day and 5day post injury timepoints. At the 1 day timepoint, we saw a significantly greater proportion of cells labeled as either SiglecF+ or Ly6G+ in young samples compared to both old and old + rMETNRL as well as slight deviation between old and old +rMETRNL (<2% mean difference) among CD45+ leukocytes and CD45+,CD11b+ myeloid cells. We interpret these findings to indicate a near equivalent degree of myeloid cells responding to injury among all groups at the three timepoints assessed but the change in the proportion of F4/80+ cells expressing high vs. low Ly-6C is statistically different at the three day timepoint (Fig 3C-E).

This quantitation does highlight the dramatic proportion of myeloid cells in the muscle at the 3 day timepoint (>60% of all cells) which is nearly halved by the 5 day measures (~30% of all cells) supporting the choice of these timepoints as intermediary and towards resolution of the myeloid response to muscle injury.

In figure 4, the three old samples are at opposite corners of the PCA plot, indicating high variability... please discuss. The methods should contain more details on the workflow used to generate the data in this figure. For example is the dendrogram/heatmap in 4d generated using all genes, or a subset? Could not find this info.

We thank the reviewer for bringing these points to our attention regarding the bulk RNA sequencing data presented in figure 4. We do believe the variability seen among the old samples is intriguing and we agree with the reviewer that this suggests highly variable gene expression patterns between the old samples; however, we interpret the clustering of young and old+metrnl samples substantiates the findings of the experiment. We have attempted to highlight this finding within the text of the results section. Furthermore, we have tried to expand upon the details of the workflow for figure 4 by expanding the text of the methods and results section. For example, we now detail that the dendrogram/heatmap shown in figure 4D is a subset of the 311 differentially expressed genes between young and old groups.

In figure 6 a few plots containing quantification of flow cytometry results are presented. The plots used for quantification should be shown with appropriate controls, ideally in the supplementary.

The reviewer correctly identifies the summary results of several flow cytometry experiments which were included in figure 6. In our revised figure 6, we have now included even more results from similar flow cytometry experiments (AnnexinV staining of muscle FAPs). We have attempted to clarify the presentation of these results by including the details on gating strategy in supplemental figure 6A and examples of FMO controls used to set gating in supplemental figure 6B.

The results reported in fig 6H are very interesting, but poorly representative of the *in vivo* situation. BMDM are not equivalent to those found in skeletal muscle, and while these experiments suggest a potential mechanism, this should be tested *in vivo*. Can etanercept block the effects of MTRNL in old animals? This should be easy to provide and it is absolutely critical.

We thank the reviewer for the interest in the results shown in figure 6H (from initial submission). My co-authors and I agree that the findings are very interesting; however, we also agree *in vivo* investigate would be best to support our *ex-vivo* results. As suggested by the reviewer, we injured 24 month old mice with BaCl₂ and administered rMETRNL in 3 doses following the injury. Following the third dose of rMETRNL (presumably once the effects of rMETRNL on the myeloid cells had initiated), we administered Etanercept *in vivo* (intraperitoneal, 1mg/kg) followed by flow cytometry and histology of muscle at day 5 following injury. Our hypothesis was that the 5 day time point would allow us to detect high levels of FAP apoptosis in old + rMETRNL treated muscle compared to old controls; a beneficial outcome for which TNF-signaling would be necessary and thus abrogated in Etanercept treated mice. Additionally, we performed Sirius Red staining of collagen to determine if altered FAP number and apoptosis were concomitant with fibrosis. We have now included a diagram of this experiment in figure 6G, representative flow cytometry results of AnnexinV staining in Fig 6H, and summary results in Fig 6I. Representative Sirius Red micrographs are shown in Fig 6J and summary quantification in 6K. Additionally, we have included example flow cytometry gating strategy/ancestry in supplemental figure 6A along with FMO controls used for gating in supplemental figure 6B. Text detailing these *in vivo* experiments was added to the animal section of the methods, the results section as well as the figure legend to figure 6. We are hopeful that inclusion of these data alleviates the concern by the reviewer about the issues raised regarding our *ex vivo* experimental system.

Reviewer #2 (Remarks to the Author):

The present work is focused on the role of Meteorin-like (Metrl) in muscle regeneration during ageing. The authors convincingly showed that Metrl decreased in aged muscle after injuries. By transfecting muscles with AAV-virus expressing Metrl it was sufficient to restore grip strength and time to fatigue after eccentric contraction damage in aged mice. Then the authors used parabiosis, bone marrow transplant and Metrl injection to further prove the role of Metrl in muscle rejuvenation. The ability of Metrl to rescue muscle regeneration was also confirmed by using Metrl knockout mice, which fails to rescue the regenerative capacity of old mice in parabiosis or bone marrow transplant experiments. To identify the insights that justify Metrl action, the authors performed scRNA seq and found changes in a subpopulation of FAPs, which become more fibrogenic in old mice but not in Metrl-treated aged animals. Finally they identify TNFa as the culprit for the persistence of the

pro-fibrogenic FAPs in aged mice. In fact, *Metrn1* restores TNF α expression in macrophages causing apoptotic cell death of FAPs and therefore reducing the pro-fibrogenic program. I like the elegant experimental design that the authors used to prove the role of *Metrn1* in muscle regeneration during ageing. The data are convincing and sustain this correlation. However, the drawback is that the insights that explain such beneficial effect are weak. The authors should consider the following points.

Point1. The authors claimed that macrophages are the major source of *Metrn1* during injury. However, this should be proven. It is strange that the peak of *Metrn1* expression happens at day 1 after BaCl₂ injection, a time in which the most abundant cells are polymorphonuclear neutrophils and not monocytes/macrophages. Please use scRNAseq or multiple FISH to prove that after eccentric contraction or BaCl₂ injection, macrophages are the only cell expressing *Metrn1*.

We agree with the reviewer, as our initial claims suggested “age-related *Metrn1* insufficiency was specifically a result of reduced macrophage expression of *Metrn1*.” were not well supported by our data. Moreover, the reviewer’s inquiry about neutrophils being a source of METNRL is certainly warranted considering the highest induction of METNRL expression is early in the regenerative time course.

In an effort to further elucidate the cell-type specific expression pattern of *Metrn1*, we performed scRNA-sequencing analysis as suggested by the reviewer. We used an integrated, large scale dataset of single cell transcriptomic signatures of muscle regeneration across several timepoints and biological ages. These data were previously published and made available by the Cosgrove lab (Commun Biol 2021; PMID 34773081) and include >365,000 individual cells across the mouse lifespan and a complete regeneration time course following muscle injury. We validated the cell identities by gene expression which matched the published metadata classifications and analyzed the expression patterns of *Metrn1* at the single cell/single nucleus level. The result of these experiments are now shown in supplemental figure 1D-G which will clearly show highly enriched expression among various immune cell subpopulations compared to all other cell types (SFig 1D,E). The expression patterns within the immune cell subpopulations match well with our original qPCR data which show a robust induction of *Metrn1* immediately following injury and tapering off later (SFig 1F). Similarly, the age-related expression of *Metrn1* matches our qPCR data where we validated the expression in 3 month and 24 month old mice, whereas this scRNAseq data reveals a similar tapering near 20months of age that extends to 30months thought. The presentation of these data using Violin plots is somewhat skewed due to changes in the number of cells included in the analysis (SFig 1G).

With these new analyses included in this revision, we want to ensure our conclusions are grounded in the data presented; therefore, we have attempted to clarify the text of the manuscript to highlight the expression of *Metrn1* across myeloid populations – not exclusively macrophages. We believe these data lend weight to this finding and may help to clarify the concerns raised by the reviewer about the claim that macrophages specifically were the source of *Metrn1*.

Point2. The *Metrn1* effect on CSA must be better characterized. Is this due to increased myoblast proliferation and fusion or to an enhancement of protein synthesis in regenerating myofibers? Both options are sustained by the presence of IGF1 and TGF β cytokines among the top differentially expressed genes in Fig 5F and both are well known modulators of protein synthesis and myogenic differentiation.

The reviewer raises an important question of enhancements to myofiber protein synthesis and myoblast proliferation due to METRNL. In fact, we have evidence that rMETRNL treatment *in vitro* shows no direct effects on proliferation of muscle stem cells measured by EdU incorporation (now included in SFig 5E). In our flow cytometry experiments where MuSCs were analyzed (CD45-, CD11b-, Ter119-, CD31-, Sca1-, VCAM1+ cells), we found no significant differences in the total number of MuSCs in old vs old+rMETRNL samples (new data added to SFig 5F). These findings would suggest that the impacts of rMETRNL on MuSCs of aged mice are secondary to the effects on the immune cell profile. To substantiate this evidence, we have included data from these experiments in supplemental figure 5 where we show the absence of any direct cellular phenotypes in primary FAPs and MuSCs following treatment with rMETRNL. We believe this revision will contribute to the overall findings by showing the effects METRNL has on myofibers or muscle stem cells is secondary to effects on immune cells.

In fact, the differentially expressed signaling pathways the reviewer points out (Fig 5F) are pathways upregulated in our scRNA-seq datasets between myeloid cells and FAPs and exclude myonuclei and MuSC populations (See also, Fig 5C,D, E). These cell subsets were chosen because FAPs contained the greatest inferred number and weight of ligand/receptor gene sets when the immune cells were chosen as the ligand source (Fig 5C,D). The receptor/ligand gene sets were far fewer and far weaker when the receiving cells were myonuclei or muscle stem cells.

Point3. Grip strength is a modest indicator of muscle function because it can be affected by pain and other inputs. Please measure muscle force by ex-vivo or in vivo experiments. Muscle force should be also tested in Fig3 experiments.

The reviewer raises valid concerns over the use of grip strength as a functional outcome of strength. We included the measurement of grip strength in our initial studies in the downhill running with AAV-*Metrn1* model (Fig1d-1h) to allow repeated measures over time. Once we moved towards the approach of intramuscular delivery of rMETRNL (Fig 3 – Fig 6), we lack any validation that rMETRNL treatment affects muscle function, as pointed out by the reviewer. In this revised submission, we have performed additional experiments using the same rMETRNL delivery strategy with the outcome measure of muscle force measured by *in situ* force production of the Tibialis anterior. Details of this method are now included in the methods section and the results are presented in 3H-J. This approach has the benefit of circumventing the weaknesses of grip strength because it is not affected by pain or mouse handling. We have done this in an effort to bolster the rejuvenation findings of rMETRNL during aged muscle regeneration shown in Figure 3 alongside the cross-sectional area data from our prior submission. We

believe inclusion of this suggested experiment by the reviewer has greatly improved the validity of our findings, by not only showing the histological changes (CSA) but now coupling these data with improved force generation (Fig 3H-J).

Point4. The connection with FAPs and TNF α expression is weak. In case that impairment of TNF α would affect FAPs survival, why young and old animals show the same level of Sca1+ cells (Fig 6e) despite the young mice show higher level of TNF α secretion (Fig 6d)? If this idea is true they should mirror the old+ *Metrn1* mice. Indeed, they show an increased cell death (Fig6f) compared to old mice.

My coauthors and I do agree with the reviewer in suggesting that the connection between FAP apoptosis and TNF could have been stronger in our original manuscript submission. We have attempted to bolster the data in support of this by performing additional, *in vivo*, experiments to assess FAP apoptosis in old muscle treated with rMETRNL and the TNF inhibitor Etanercept (now Fig 6g). As per the disconnection between FAP number and TNF α expression, we agree, the finding that a lower amount of TNF secreted by myeloid cells 3 days post-injury does not match with the number of FAPs 7 days post-injury in the old control group does raise questions. It is our thinking that this discrepancy could be due to the long duration between measurements (3days vs 7days). It is also possible that FAPs in the old control samples have begun the differentiation process towards a pro-fibrotic phenotype (Fig 6B) in which case they could become more resistant to apoptosis-induction by TNF α (in agreement with Tidball & Henricks, 2015; Lemos et al., 2015). We hope our new data (Fig 6) and discussion here help clarify the discrepancies mentioned by the reviewer.

Point5. How TNF α is modulated by *Metrn1*? Which pathway is involved?

The Reviewer highlights an interesting question that our initial submission leaves unanswered: How is TNF α modulated by METRNL? Our group has previously published data which details the regenerative characteristics of muscle in whole body *Metrn1*-KO mice. This publication shows no changes in *Tnfa* mRNA expression in muscle extracts 1 day following injury in young, *Metrn1*-KO vs WT mice; however, *Metrn1*-KO mice have greater *Tnfa* mRNA in muscle 4 days following injury (Nature Metabolism 2020; PMID: 32694832; Figure 1). This data could suggest that METRNL negatively regulates *Tnfa* mRNA although not during the initial stage of injury. We have also seen that in young, c57b6 mice given an AAV-*Metrn1* vector, there is greater mRNA for *Tnfa* in the muscle following injury suggesting positive regulation of *Tnfa* by METRNL (data not shown). In the current manuscript, we elected to use ELISA on media conditioned by primary CD45+, CD11b+ isolated 3-days post injury to directly assess the levels of TNF- α protein that is secreted. We believed this approach would be superior to measurement by qPCR to account for altered regulation at the secretory stage. These results (Fig 6C) suggest that TNF α secretion is lowered in the isolated myeloid cells from aged mice at the 3-day post-injury timepoint (concomitant with reduced *Metrn1* mRNA) and that treatment with rMETRNL upregulates TNF α secretion in isolated, old myeloid cells. Taken together our data does not support any specific pathway by which METRNL contributes to TNF α modulation. There may be

discrepancies between the young vs old response and whether TNF is measured at the mRNA or protein level. We cannot at this time determine any specific pathways responsible for the METRNL-induced TNF alterations. Recently published evidence suggest METRNL acts as a ligand for a KIT receptor in the myocardium (PMID: 35709278); however, we have not independently verified these results nor does this immediately suggest a mechanism for canonical TNF induction by myeloid cells.

With these points in mind, we have rewritten portions of the discussion section to include this background/context and to address the question of TNF α modulation by Metrnl.

Point6. Fig 7g-h suggest a potential role of TNF α pathway in regulation of pro-fibrogenic FAPs but does not show that it is the culprit of the muscle phenotype in aged mice. To sustain the authors hypothesis, authors must prove *in vivo* that: 1) the profibrogenic FAPs are indeed responsible for the decreased CSA and muscle force in BaCl₂ treated mice and after eccentric contraction, 2) the TNF α expression of macrophages affect the FAPs subpopulation, the decreased CSA and muscle force in BaCl₂ treated mice and after eccentric contraction.

The point raised by the reviewer referring to figure 6g,h (from the initial submission) is well received; in fact, the conclusion we were hoping to highlight in figure 6 is exactly as the reviewer mentions: that the TNF α pathway regulates pro-fibrogenic FAPs. Indeed this pathway has previously been shown to modulate FAPs and fibrosis in the DMD mouse model (Lemos et al., 2015). Our goal was to work within this mechanism and focus on TNF α as a mechanism by which rMETRNL contributes to less FAP fibrogenic differentiation and greater FAP clearance in the context of aging muscle. To this end, we have performed additional *in vivo* experiments as suggested by the reviewer. We used rMETRNL treatment and the TNF inhibitor Etanercept to show 1-the role of TNF in METRNL-driven FAP apoptosis (Fig6g-i) and 2- its regulation on muscle fibrosis (Fig6j,k), all *in vivo*. The addition of these experiments in the revised manuscript greatly supports a role for TNF α regulation of pro-fibrogenic FAPs *in vivo* in aged mice.

In addition to the experiments described above (Fig 6), we now include *in-situ* force measurements in Figure 3 showing the benefits of rMETRNL to enhance force production after injury in aged muscle, which may also ease concerns raised by the reviewer in Point 6.

REVIEWERS' COMMENTS

Reviewer #1 (Remarks to the Author):

I must commend the authors for addressing all my main criticism with well designed experiments. I have no further concern.

Reviewer #2 (Remarks to the Author):

The authors addressed most of my concerns. The paper is improved